# Rethinking the Architecture Design for Efficient Generic Event Boundary Detection

## ABSTRACT

Generic event boundary detection (GEBD), inspired by human visual cognitive behaviors of consistently segmenting videos into meaningful temporal chunks, finds utility in various applications such as video editing and summarization. In this paper, we demonstrate that state-of-the-art GEBD models often prioritize final performance over model complexity, resulting in low inference speed and hindering efficient deployment in real-world scenarios. We contribute to addressing this challenge by experimentally reexamining the architecture of GEBD models and uncovering several surprising findings. Firstly, we reveal that a concise GEBD baseline model already achieves promising performance without any sophisticated design. Secondly, we find that the common design of GEBD models using image-domain backbones can contain plenty of architecture redundancy, motivating us to gradually "modernize" each component to enhance efficiency. Thirdly, we show that the GEBD models using image-domain backbones conducting the spatiotemporal learning in a spatial-then-temporal greedy manner can suffer from a distraction issue, which might be the inefficient villain for the GEBD. Using a video-domain backbone to jointly conduct spatiotemporal modeling for GEBD is an effective solution for this issue. The outcome of our exploration is a family of GEBD models, named EfficientGEBD, significantly outperforms the previous SOTA methods by up to 1.7% performance growth and 280% practical speedup under the same backbone choice. Our research prompts the community to design modern GEBD methods with the consideration of model complexity, particularly in resource-aware applications. The code is available at https://github.com/anonymous.

## CCS CONCEPTS

• **Computing methodologies** → **Video segmentation**; **Activity recognition and understanding**.

## KEYWORDS

Generic event boundary detection, Video understanding

## 1 INTRODUCTION

Video understanding has gained significant traction in multimedia fields recently [10, 34, 53]. Motivated by cognitive science

**Unpublished working draft. Not for distribution.**

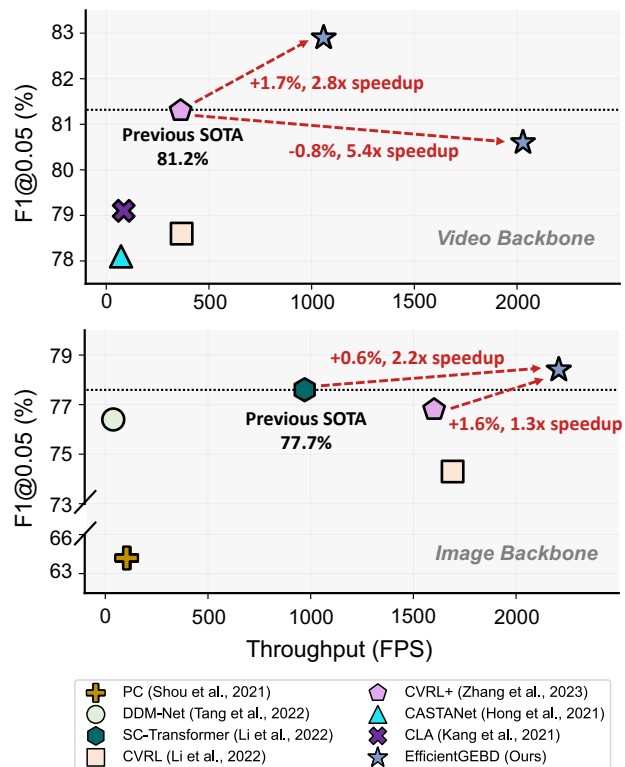

**Figure 1: The throughput (frames per second, FPS) *vs.* F1 score of different methods using video and image domain backbones on Kinetics-GEBD [41].**

showing human naturally tends to parse long videos into meaningful segments for subsequent comprehending [36], recent researchers have proposed the Generic Event Boundary Detection task (GEBD) [41] that aims at detecting the moments in videos as generic and taxonomy-free event boundaries. Generally, the development of GEBD task will be valuable in immediately supporting applications like video editing [5] and summarization [16], and more importantly, spurring progress in long-form video, where GEBD can be implemented as the first step towards segmenting video into units for further reasoning and understanding.

The potential value of GEBD has promoted the development of video GEBD benchmark competition, such as LOVEU21-23 [40, 42, 43]. However, the requirements of such competitions tend to encourage the models to achieve high performance without considering model complexity. From Figure 1, we see that some of the existing GEBD methods [17, 23, 41, 47] indeed suffer from the efficiency issue and have a low throughput, which can hinder the real-world applications of GEBD. On one hand, efficiency itself is supposed to be an important evaluation metric for any GEBD method. On the other hand, as GEBD can serve as a pre-processing step, its

high latency will surely lead to the inefficiency of understanding of long-form videos.

This paper rethinks the design paradigm of the deep network-based GEBD model and is intended to improve its efficiency. We surprisingly find that a basic GEBD model containing the most concise design can already achieve comparable detection performance (F1 score of 77.1%) with the soft-label training techniques in [28, 52]. We call this basic GEBD model as BasicGEBD. Considering that early GEBD methods [41, 47] do not apply soft-label during training, we imply a portion of the detection performance differences between existing GEBD methods may be due to these training techniques.

We then use the BasicGEBD as the starting point to gradually "modernize" each component of BasicGEBD towards a highly efficient GEBD model. Our investigation of the backbone reveals that the commonly applied ResNet50 [15] contains plenty of redundancy and increasing the capacity of the backbone in BasicGEBD is no benefit to GEBD performance. This motivates us to reduce the size of the backbone to improve efficiency. We then reexamine the rest components and find that lightweight designs can effectively achieve temporal modeling for GEBD. Specifically, we build the encoder based on difference maps [47] capturing the local relation and use a small convolution network to process the similarity matrix to extract the global information. Then we further propose to use a cross-attention module to fuse global-local information for final predictions. We show that each modified component is effective yet lightweight compared to previous methods. Finally, we obtain a family of GEBD models named EfficientGEBD, which can achieve a new SOTA result (78.3%) on Kinetics-GEBD while with 2.2× speedup than the previous SOTA methods [28] using the same backbone (Figure 1).

Then we naturally think about why using high-capacity backbone models does not benefit the final GEBD performance. Our results show that using image-domain backbones conducting the spatiotemporal learning in a spatial-then-temporal greedy manner can lead to a *Distraction* issue, which can distract the attention of the backbone from the true boundary-related objects. Such an issue can be due to the absence of temporal modeling ability of the image-domain backbone and therefore will be effectively addressed by implementing a video-domain backbone to jointly conduct spatiotemporal modeling for GEBD, which further boosts the performance of EfficientGEBD by a large margin.

We hope the new observations and discussions can challenge some common designs and existing evaluation metrics in GEBD tasks. First, we suggest considering efficiency as an important metric for evaluation to ensure the applicability of GEBD models. Second, using small image-domain backbones is sufficient for GEBD models which improves the efficiency with high detection performance. Third, developing a GEBD model directly based on the video-domain backbone is suggested for future works. Our contributions can be summarized as follows:

- We introduce a strong baseline model for GEBD tasks, BasicGEBD, which achieve high performance without any sophisticated designs.
- By detailed studying each component of BasicGEBD, we obtain a family of GEBD models, named EfficientGEBD, achieving SOTA performance with high inference speed.

- Implementing video-domain backbone for spatiotemporal modeling can significantly boost the performance of EfficientGEBD (82.9%) and even BasicGEBD (82.5%).
- Extensive experiments and studies on the Kinetics-GEBD [41] and TAPOS [38] datasets provide new experimental evidence as well as demonstrate the effectiveness and efficiency of our work on GEBD tasks.

## 2 METHODOLOGY

In this section, we provide a trajectory going from a basic GEBD model to an efficient GEBD model. To start the network designing, we first build a baseline model, named BasicGEBD, as a starting point by abstracting existing popular GEBD methods [24, 28, 41, 47]. By reexamining each component of BasicGEBD, we propose to reduce the surplus computational costs and improve the model efficiency, resulting in a new family of efficient GEBD models, named EfficientGEBD. Moreover, by investigating the possible reason why the larger size of the backbone model turns into surplus costs rather than improving GEBD performance, we are motivated to jointly conduct spatiotemporal modeling in the backbone by using a video domain deep network when building GEBD models. In Figure 2, we show the procedure and the results we are able to achieve with each step of the "model modernization". All models in Section 2 are trained and evaluated on Kinetics-GEBD [41], a challenging and well-known GEBD dataset, which will be introduced in Section 3.

**Training settings.** Due to the similar duration and FPS of all videos in Kinetics-GEBD, we uniformly sample 100 frames (i.e., $T = 100$) from each video as inputs. The length of a video clip is set to 17, where the model is designed to recognize whether the median frame is the boundary or not. We apply the training settings in [28, 52] that use Gaussian Smoothing in all the following studies. The whole model is trained end-to-end for 15 epochs with Adam [25]. The learning rate is set as 1e-2, which will be divided by 10 at the 6th and 8th epochs, respectively. We will use this fixed training recipe with the same hyper-parameters throughout this section. Here, we briefly introduce the Gaussian Smoothing:

According to [28], Gaussian Smoothing proposes to smooth the sparse one-hot labels, $Y \in \{0, 1\}^T$, with a Gaussian kernel with the width of $\sigma$ to generate soft labels $\widetilde{Y} \in [0, 1]^T$, where $T$ is the length of video. The smoothed label can provide more boundary information and is effective in improving detection performance. We use $\sigma = 1$ for all our experiments.

**Evaluation Protocol.** As mentioned in [41], the Relative Distance (Rel.Dis.), which represents the error between the detected and ground truth timestamps divided by the length of the corresponding action instance, is used to determine whether detection is correct or not. Following [40, 42, 43], we report the F1 scores with the Rel.Dis. of 0.05 in this section. Moreover, theoretical (GFLOPs) and practical (FPS) speeds are used for efficiency evaluation. The throughput is tested using the maximal batch size for each model running on a NVIDIA RTX 4090 GPU with mixed precision.

## 2.1 A baseline model for GEBD: BasicGEBD

Although the design paradigms, such as unsupervised or self supervised manners, can also be used for building GEBD methods, in this

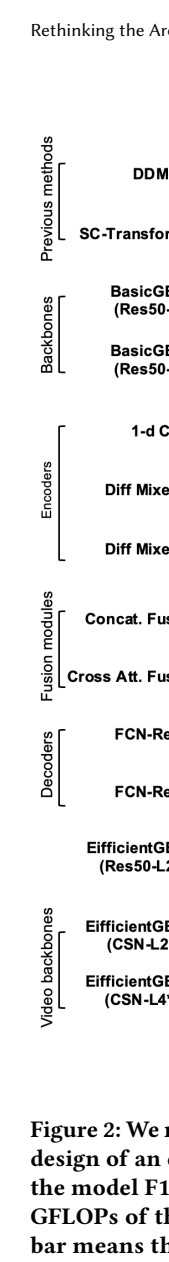

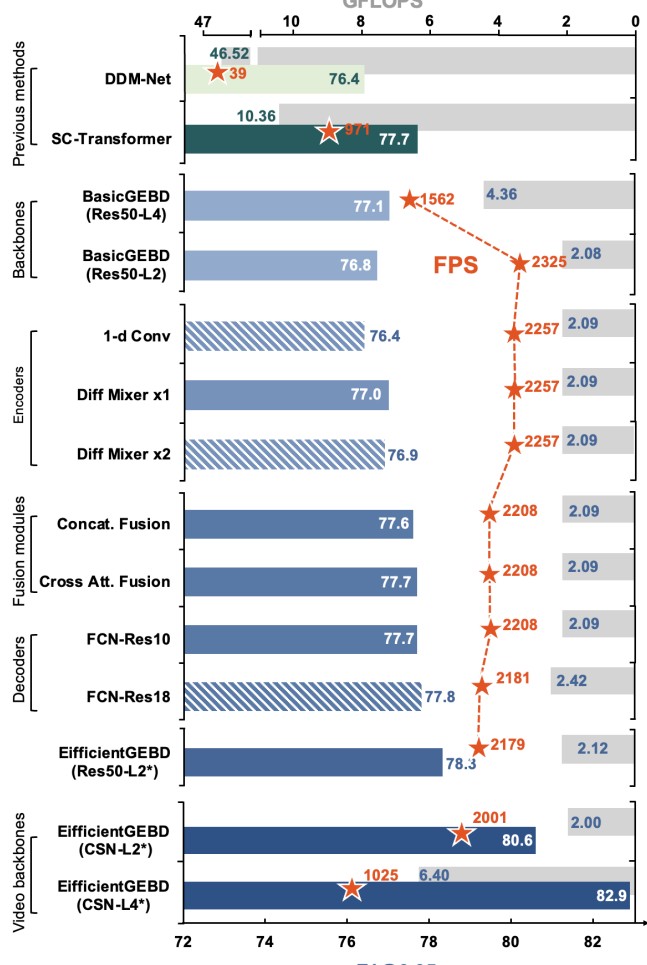

Figure 2: We modernize the proposed BasicGEBD towards the design of an efficient GEBD model. The foreground bars are the model F1@0.05 score of the detection performance. The GFLOPs of the model are depicted by grey bars. A hatched bar means the modification is not adopted. In the end, our EfficientGEBD with ResNet50-L2* can outperform the previous SOTA method (SC-Transformer [28]), and can be further obviously improved by using CSN backbone [49].

paper, we mainly focus on the GEBD models that treat the boundary detection tasks as supervised video clip binary classification tasks, which are trained in an end-to-end manner. Other types, such as detection-based GEBD models in an off-line feature extraction manner (Temporal Perceiver [46]), are not included in our study. SBoCo [24], DDM-Net [47], and SC-Transformer [28] (SC-Trans.) are selected as the representative supervised GEBD networks. By studying their architectures, we show that these models follow the five components design paradigm: (1) The backbone for feature extraction; (2) The encoder for temporal modeling; (3) The similarity map (Sim. Map); (4) The decoder processing the similarity map; (5) The feature fusion module. These components are summarized in

Table 1. Generally, these methods conduct the spatiotemporal modeling in a greedy step-by-step manner, where the backbone is first used to extract the spatial representations and then the temporal modeling is conducted by the subsequent modules. More details can be found in our Supplementary materials.

As a baseline model, we use the most concise design for each component: (1) The widely used ResNet50 [15] pre-trained on ImageNet [6] is applied as the backbone. (2) A 1-d Conv layer, consisting of BN [22], Conv (Kernel=3) and ReLU, is implemented as the encoder. (3) The cosine similarity (CosSim.) is used to generate the similarity matrix. (4) A fully convolutional network (FCN), which normally consists of a mini ResNet10[1] following previous designs [29, 47, 52] as the decoder. (5) As only DDM-Net applies the fusion module, we do not use it in our baseline model. The obtained architecture is shown in Figure 3 and we call this basic GEBD model BasicGEBD in our research.

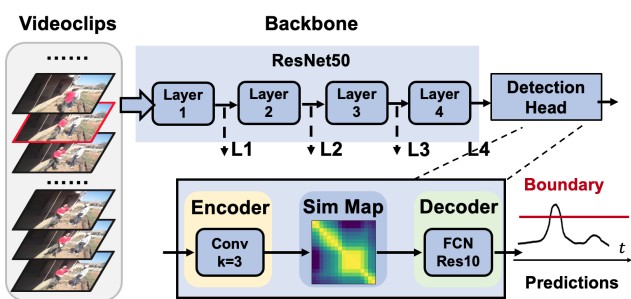

Figure 3: The architecture of BasicGEBD.

Surprisingly, we found that the performance of the proposed BasicGEBD can be up to 77.1% in the term of F1@0.05 with the training settings used in [28, 52], which is already superior to most existing GEBD methods (shown in Table 2). It is amazing that such a concise model can achieve high detection results without any additionally sophisticated architecture designs as well as also show high efficiency in terms of both theoretical (FLOPs) and practical (Throughput) speed (shown in Figure 2 and Table 1). From the results, we conclude that apart from the design of the GEBD network architecture, the training procedure also affects the ultimate performance of GEBD models. Therefore, we infer that the superior performance of some modern GEBD models actually benefits not only from the sophisticated architecture design but also from the advanced Gaussian smoothing training techniques.

*We then use the BasicGEBD as the start point to explore a more efficient model for GEBD tasks.*

## 2.2 Exploring for EfficientGEBD

Our exploration from BasicGEBD to EfficientGEBD is by gradually "modernizing" the BasicGEBD from four aspects: 1) backbone network; 2) encoder; 3) decoder; 4) fusion module, where the backbone is for spatial modeling, and the rest are designed for temporal modeling. In Figure 2, we show this procedure and the results we are able to achieve with each step of the model modernization.

---

[1]A ResNet10 has 4 residual layers, where each layer contains 1 residual block, consisting of two convolution layers with residual connection.

**Table 1: The architectures of three representative GEBD methods and the propose models in this paper.**

| Mehtods | Backbone (FLOPs) | Encoder (FLOPs) | Sim. Map | Decoder (FLOPs) | Fusion (FLOPs) | F1@0.05 | FLOPs |
|---------|------------------|-----------------|----------|-----------------|----------------|---------|-------|
| SboCo [24]† | ResNet50 | 1d-Conv | CosSim. or L2-Sim. | ResNet + Transformer | - | 73.2 | 163.5G |
| DDM-Net [47] | ResNet50 (4.10G) | Differences (0.0G) | L2-Sim. | FCN (0.25G) | Progressive Att. (1.02G) | 76.4 | 46.52G |
| SC-Trans. [28] | ResNet50 (4.10G) | Transformer (60M) | CosSim. | FCN (5.92G) | - | 77.7 | 10.36G |
| BasicGEBD | ResNet50-L4 (4.10G) | 1d-Conv (0.04M) | CosSim. | FCN (0.25G) | - | 77.1 | 4.36G |
| EfficientGEBD | ResNet50-L2* (1.82G) | DiffMixer (17.9M) | CosSim. | FCN (0.25G) | Cross Att. (0.8M) | 78.3 | 2.12G |
| EfficientGEBD | CSN-L2* (1.72G) | DiffMixer (17.9M) | CosSim. | FCN (0.25G) | Cross Att. (0.8M) | 80.6 | 2.00G |
| EfficientGEBD | CSN-L4* (6.09G) | DiffMixer (26.8M) | CosSim. | FCN (0.25G) | Cross Att. (0.8M) | 82.9 | 6.40G |

† The FLOPs of each component can not be calculated since the official code is not available. The overall FLOPs are referred from [13].

2.2.1 *Backbone network.* From Table 1, we see that the backbones (ResNet50) generally have a large contribution to the whole computational costs for both BasicGEBD and other existing GEBD methods, which motivates us to investigate how the size of the backbone model can affect the performance of a GEBD model.

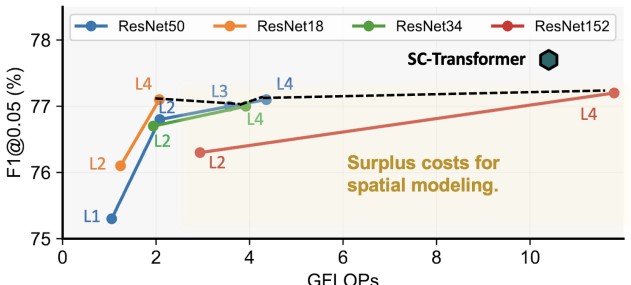

**Figure 4: The GFLOPs *v.s* F1 score of BasicGEBD with different sizes of ResNets as the backbone.**

Therefore, we implement the BasicGEBD with the ResNets with different capacities, and the results are shown in Figure 4. From the results, we see that using a larger backbone network seems to be unnecessary for performance improvement: the BasicGEBD with ResNet18 can already achieve the detection accuracy compared to that with ResNet50, indicating the extra computational costs of ResNet50 can be superfluous. This can be further confirmed by the results of BasicGEBD with ResNet152. This finding contradicts the general common sense in other vision tasks, such as object detection [32, 33], where using a larger backbone model normally brings obvious performance improvements. Considering that the ResNets are used for spatial modeling in these GEBD models, we hypothesize that there exist tons of redundant computational costs for spatial modeling when we use ResNet50 or larger models as the backbone in BasicGEBD. Such a hypothesis is further confirmed by the high performance achieved by the BasicGEBD when we attach the detection head at early layers of ResNet50 (E.g, ResNet50-L2), where the model achieves valuable detection performance (76.8%), which is already superior to most of the existing GEBD methods (see the results in Table 2). The BasicGEBD with ResNet50-L2 also has much smaller computational costs (2.08 GFLOPs), which is over 45% smaller than the original BasicGEBD, approximately 5 times smaller than SC-Transformer [28] (10.36 GFLOPs) and over 20 times smaller

than DDM-Net [47] (40 GFLOPs) (shown in Figure 2). The lower complexity leads to the fast throughput of the modified GEBD model (2,325 FPS). We also found that the performance of BasicGEBD can drastically drop with too small a capacity of backbone, indicating that the capacity of the backbone for GEBD should be large enough for spatial modeling.

As the surplus costs for spatial modeling exist when we use ResNet50 as the backbone, we can make use of such redundancy to improve the efficiency of our model with barely any performance loss, motivating us to reduce the size of the backbone. For a fair comparison with previous research such as [28, 47] that all use ResNet50 as the backbone, we use ResNet50-L2 to conduct the following examinations. However, our findings and designs also hold for other small-size backbone, such as ResNet18 and ResNet34, which will be shown in the experiments.

*Then, we will use the first two layers of the backbone network (ResNet50-L2) to build the efficient GEBD model.*

2.2.2 *Temporal modeling ability in encoder.* As we have achieved the upper limit of spatial modeling, we then need to increase the temporal ability for accurate boundary detection, motivating the previous researchers to use a sophisticated self-attention-based encoder [28]. However, we show that the concise design in [47] introducing difference maps for temporal modeling can achieve better performance. Such a difference design also meets our intuition in boundary detection tasks: To detect these motion-related boundaries, motion information plays a principal role in perceiving temporal variations and can be effectively modeled by using feature differences at different timestamps.

Figure 5 (a) shows the architecture of the proposed encoder, named difference mixer (Diff Mixer). Different from [47], we propose to use both difference features, $\mathbf{X}^D$, and the original features, $\mathbf{X}^I$, for temporal modeling. Given the extracted features $\mathbf{X} \in \mathcal{R}^{T \times C}$, where $T$ is the temporal length and the $C$ is the number of channels, the difference can be calculated by $\mathbf{X}_t^D = \mathbf{X}_t - \mathbf{X}_{t+1}$, $t = 1, 2, ..., T$. And we further pad $\mathbf{X}_1$ at the beginning of $\mathbf{X}^D$ to guarantee that $\mathbf{X}^D$ and $\mathbf{X}$ have the same dimension. Then $\mathbf{X}^D$ and $\mathbf{X}^I$ will be processed by a conv layer (BN-Conv-ReLU) with the kernel size of 1.

Such a design enables the encoder to effectively model the local relationship among different video frames: shot changes and motion-related boundaries can be effectively identified by the original $\mathbf{X}^I$ and difference $\mathbf{X}^D$ features, respectively, which can be

**Figure 5: The illustrations of the encoder (a) and the fusion module (d,e). In (b,c), we calculate the L2-norm and the cosine similarity map of the features at different timestamps to see whether the discriminative boundary features can be captured.**

illustrated by Figure 5 (b,c). We plot the variants of the features norm and similarity maps for $\mathbf{X}^I$ and $\mathbf{X}^D$. The shot changes can be obviously identified with the extracted features without using differences. However, for the action changes, temporal difference seems a more important cue for boundary identifying.

As depicted in Figure 2, the implementation of Diff Mixer can boost the performance of our model to 77.0% while approximately maintaining the throughput of BasicGEBD. We also further test using the Marco design of Transformers [28] as the encoder, and observe that there are no obvious improvements in the performance. Increasing the number of Diff Mixers also does not benefit the GEBD performance (the F1 score drops to 76.9% when we use two Diff Mixers). More additional encoder architecture studies can be found in our supplementary materials.

*We then use one Diff Mixer as the encoder.*

*2.2.3 Feature fusion module.* Feature fusion mechanisms which introduces the features before the decoder for final predictions, is proposed in [47] (Progressive Att.) to improve the GEBD performance. However, the computational costs of Progressive Att. can be up to 1.02G. In this section, we investigate how to conduct such a feature fusion mechanism with a lightweight design to improve the performance of GEBD.

Two fusion modules shown in Figure 5 (d,e) are tested, where the fusion can be achieved by either directly concatenating two features or using cross attention for fusion. For concatenation fusion, the features from the encoder will be firstly processed by a Conv layer (BN-Conv-ReLU) with a kernel size of 3, and then concatenated to the features from the decoder. For the cross attention (Cross Att.) fusion in Figure 5 (e), two squeeze-and-excitation (SE) [19] are used to generate the weights, which are then used to re-weight the resultant features from each branch.

The results in Figure 2 show that two fusion mechanisms can boost the performance of the modified BasicGEBD to 77.6% and 77.7%, respectively. Note that the proposed fusion model is significantly smaller than that in [47] (shown in Table 1), and therefore barely affects the efficiency. So far, the model has achieved the performance that is comparable to the previous SOTA results achieved by SC-Transformer [28] with ResNet50 backbone. However, for inference speed, the modified BasicGEBD with Cross Att. Fusion can run over 100% faster than SC-Transformer in terms of FPS.

We further indicate that such improvements can be due to the fusion of global-local temporal information. On one hand, the decoder can gather global information by using the FCN to process the similarity matrix of the whole video clip. Therefore, the features

from the decoder can be viewed as the global representations of the event boundary. On the other hand, the difference features from Diff Mixer measure the relationship for the adjacent frames (local information), which potentially captures motion cues in the temporal dimension (as it is shown in Figure 5 (b,c)), introducing them in the final predictions might immediately improve the identification of some event boundaries, and therefore benefit whole GEBD tasks.

*We then use Cross Att. fusion in our model.*

*2.2.4 The FCN in decoder.* The implementation of the similarity maps as well as the 2-d FCN decoder can be seen as a sign of recent outperforming supervised GEBD methods. The FCN normally consists of a mini ResNet (such as ResNet10 in [47], or ResNet18 in [29, 52]) or its variants (a 4-layer fully convolutional network in [28]). Such a design is important to achieve accurate detection performance for GEBD models. For example, if we apply the "old fashion" design in [41] which does not apply the similarity map and the FCN decoder (the features from encoder will be directly used for final predictions), the performance of BasicGEBD will drastically drop to 64.6% even with the DiffMixer.

From Figure 2 we see that the model with ResNet18 as the decoder is only 0.1% higher than that using ResNet10 (77.7% *v.s* 77.8%), indicating that increasing the size of the FCN does not necessarily lead to better performance. The applied decoder only has 0.25G FLOPs, which is much smaller than the decoder in [28] (5.92G). Therefore, we indicate that the large decoder in the SC-Transformer might contain too much redundancy, which can limit its efficiency. Therefore, we use the ResNet10 as the decoder for the higher efficiency of the model.

*We then use a ResNet10 as the decoder for the final model.*

*2.2.5 EfficientGEBD.* We have finished our first "playthrough" and have the general architecture of the proposed GEBD model. Moreover, following [47], we introduce both the features from layer-1 and layer-2 (L1 and L2) and concatenate them as the inputs for the encoder, which further boost the performance of the model from 77.7% to 78.3%, while hardly affects the inference speed. Here, we use ResNet50-L2* to denote that both the features from layers 1 and 2 are used. To this end, we have discovered the efficient GEBD model, named EfficientGEBD, that achieves SOTA performance compared to the existing GEBD methods using ResNet50 as the backbone on the Kinetics-GEBD dataset. Remarkably, our EfficientGEBD achieve 0.6% higher performance while having 2.2× speedup compared to the previous SOTA GEBD method, SC-Transformer [28].

Although we have explored ways to reduce the surplus computational costs for spatial modeling to build EfficientGEBD, the inefficiency issue of the backbone network is still like a dark cloud on the horizon of building high-performance GEBD models. Indeed, the results in Figure 4 show that scaling up the backbone to ResNet152 does not obviously help in performance improvement of EfficientGEBD. To achieve higher detection performance, recent studies [46, 52] and the winner solutions of GEBD competition [17, 18, 45] have proposed to implement deep models for video domain to build GEBD methods, such as Two-streams networks (TSN) [9] and channel-separated video network (CSN) [49]. The applications of CSNs built based on ResNet152 significantly boost the performance of the GEBD models in [29, 52]. As the used CSN is also built based on ResNet152, we are interested in the reason why using video backbone brings obvious improvements, which is further investigated in the following.

## 2.3 Distraction issue in GEBD

So far, all previously examined GEBD models conduct the spatial and temporal modeling in a greedy learning manner, where an image-domain backbone, such as a ResNet, is firstly used to extract the spatial features, and then the subsequent modules are applied to explore the temporal information for event boundary detection.

However, we hypothesize that conducting the spatiotemporal representation learning in such a greedy way can lead to several inefficiency issues in GEBD tasks. As the image domain backbones are usually designed to identify the main objects in an image, learning the spatial features without the guide from temporal information can result in the attention of the backbone distracting from the objects most related to the boundaries, and getting stuck in some areas containing the other objects in each frame. We refer to such an issue as *distraction issue*. Figure 6 further illustrates the distraction issue when we use the image-domain backbone to build GEBD models. For instance, the event boundary in Figure 6 (a) is defined by the changes of action during arm wrestling. The high activations of the ResNet backbone get stuck in the spatial areas that contain the head of the person in the center of the frame. With the arm wrestling-related spatial areas missing features extracted from the backbone, the subsequent modules will have difficulties conducting the following temporal modeling, resulting in the failure detection of this boundary. More experimental visualization results are provided in supplementary materials. Therefore, we indicate the distraction issue may be the real villain for the inefficiency when we use image domain backbones to build GEBD models.

However, from Figure 6, we see that high activations of CSN surround the arm wrestling-related spatial areas. Actually, spatial and temporal learning can complement each other during feature learning. On one hand, temporal information can be important to guide the backbone to focus on boundary-related spatial representations. On the other hand, the learned spatial features can be used to explore the temporal variants, which are important cues for boundary detection. Therefore, instead of learning spatiotemporal features in a step-by-step manner, a more reasonable way is to jointly extract the spatiotemporal features in the GEBD backbone by using video-domain backbone networks, which effectively avoids the aforementioned static trap and improves the GEBD performance.

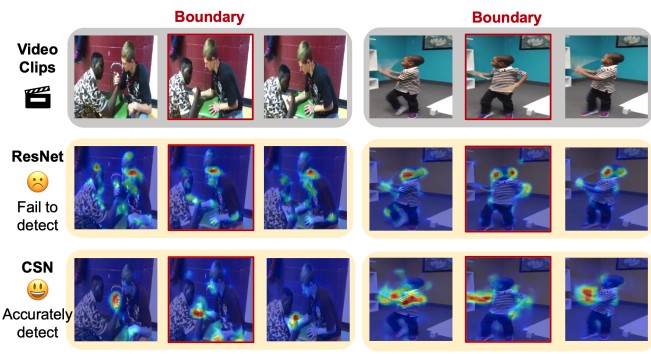

(a) Change of arm direction in wrestling.    (b) Change of swing direction.

**Figure 6: The activations captured by GradCAM++ [3] using ResNet [15] and CSN [49] as the backbones for GEBD models. The median frame is the boundary frame. The yellow circles show that the CSN focuses more on the spatial areas that related to the actions rather than the static object.**

To this end, channel-separated video network (CSN) [49], pre-trained on video action recognition datasets, IG65M [11], is implemented as the backbone to improve BasicGEBD and EfficientGEBD. By using CSN-L4 as the backbone model, the BasicGEBD surprisingly achieves the performance of 82.5% (see Table 2), which outperforms all previous published researches [29, 46, 52]. Moreover, the EfficientGEBD with CSN-L2 and CSN-L4 can achieve the performance of 80.4% and 82.0%, respectively (these results are reported in supplementary materials). Compared to the scenario that introducing additional model capacity does not bring obvious GEBD performance when we use an image-domain backbone, the superiority of CSN-L4 compared to CSN-L2 indicates that scaling behavior does exist when we use video-domain backbone network for GEBD models, which demonstrates the importance of conducting spatiotemporal modeling in the backbone model to build high-performance GEBD models.

As depicted in Figure 2, EfficientGEBD with CSN-L2* and CSN-L4* achieve the performance of 80.6% and 82.9%. However, our simple yet effective models are still highly efficient, EfficientGEBD with CSN-L2* achieving a throughput of over 2,000, which is over 4× faster than previous SOTA result in [52]. When using CSN-L4*, we can achieve a new SOTA result of 82.9% on Kinetics-GEBD with over 1,000 FPS, which is 180% faster than [52]. These encouraging findings prompt us to rethink the correctness of using an image domain backbone when building GEBD models.

## 3 EXPERIMENT

### 3.1 Experimental Settings

*3.1.1 Datasets.* The empirical evaluations are based on the frequently used Kinetics-GEBD [41] and TAPOS [38] datasets following most previous works [24, 29, 41, 47]. The Kinetics-GEBD dataset contains 54,691 videos randomly selected from Kinetics-400 [2], and these videos are labeled with 1,290,000 generic event temporal boundaries. The distribution of videos across training, validation, and testing sets in Kinetics-GEBD is nearly uniform, maintaining a ratio close to 1:1:1. Each video is annotated by five annotators, resulting in an average of approximately 4.77 boundaries per video.

**Table 2: Comparisons in terms o F1 score (%) on Kinetics-GEBD with Rel.Dis. threshold from 0.05 to 0.5.**

| Method | Backbone | F1 @ Rel. Dis. | | | | |
|--------|----------|------|-----|-----|-----|-----|
| | | 0.05 | 0.1 | 0.3 | 0.5 | avg |
| BMN [31] | Res50 | 18.6 | 20.4 | 23.0 | 24.1 | 22.3 |
| BMN-StartEnd [31] | Res50 | 49.1 | 58.9 | 66.8 | 68.3 | 64.0 |
| TCN-TAPOS [26] | Res50 | 46.4 | 56.0 | 65.9 | 68.7 | 62.7 |
| TCN [26] | Res50 | 58.8 | 65.7 | 70.3 | 71.2 | 68.5 |
| PC [41] | Res50 | 62.5 | 75.8 | 85.3 | 87.0 | 81.7 |
| PC+OF [41] | Res50 | 64.6 | 77.6 | 86.4 | 87.9 | 83.0 |
| SBoCo [24] | Res50 | 73.2 | - | - | - | 86.6 |
| Temporal Per. [46] | Res50 | 74.8 | 82.8 | 87.9 | 89.2 | 86.0 |
| CVRL [29] | Res50 | 74.3 | 83.0 | 88.6 | 89.8 | 86.5 |
| CVRL+ [52] | Res50 | 76.8 | 84.8 | 89.6 | 90.6 | 87.7 |
| DDM-Net [47] | Res50 | 76.4 | 84.3 | 89.2 | 90.2 | 87.3 |
| SC-Transformer [28] | Res50 | 77.7 | 84.9 | 90.0 | 91.1 | 88.1 |
| **BasicGEBD** | Res50 | 76.8 | 83.4 | 88.5 | 89.6 | 86.6 |
| **EfficientGEBD** | Res50-L2* | **78.3** | **85.1** | **90.1** | **91.3** | **88.3** |
| SBoCo [24] | TSN | 78.7 | - | - | - | 89.2 |
| CLA [23] | TSN | 79.1 | - | - | - | - |
| CASTANet [17] | CSN | 78.1 | - | - | - | - |
| CVRL [29] | CSN | 78.6 | - | - | - | - |
| CVRL+ [52] | CSN | 81.2 | - | - | - | - |
| **BasicGEBD** | CSN | 82.5 | 87.7 | 91.9 | 92.8 | 90.4 |
| **EfficientGEBD** | CSN | **82.9** | **88.2** | **92.2** | **93.2** | **90.8** |

Since the annotations for the test set are not publicly accessible, we conducted training on the training set and subsequently evaluated model performance using the validation set. In line with the methodology outlined in [41], we adapt TAPOS by concealing each action instance's label and conducting experiments based on this modified dataset.

*3.1.2 Implementation Details.* For Kinetics-GEBD, we use the same implementation details as we described in Section 2.2. As the videos in TAPOS have a large variety of duration over instances, we split the instances without overlapping and sample 100 frames by keeping a similar FPS to Kinetics-GEBD following [46]. The scores of sub-instances are merged together to generate the final prediction. The whole model is trained end-to-end for 30 epochs with Adam [25] and a base learning rate of 2e-2, which will be divided by 10 at the 6th, 8th and 15th epochs, respectively. Other settings are the same as these for Kinetics-GEBD.

## 3.2 Empirical Evaluations on Kinetics-GEBD

Table 2 and Figure 7 show the results of our models on the Kinetics-GEBD validation set, with Rel.Dis. threshold from 0.05 to 0.5. The complete results are provided in supplementary materials. Overall, we see that our methods achieve promising performance compared to previous methods in different Rel. Dis.. Moreover, we find that EfficientGEBD is superior to previous GEBD methods, especially under the most stringent Rel. Dis. constraint (0.05), indicating the stronger boundary detection ability of our method. Specifically in terms of F1@0.05. Specifically, when using the ResNet50 as the backbone, EfficientGEBD competes SC-Transformer [28] with 0.6% higher performance (78.3% *v.s* 77.7%) and 2.2× speedup (2208 FPS *v.s* 971 FPS). EfficientGEBD achieves new SOTA performance of

82.9% when using CSN as backbone model, which is over 1.7% higher and 2.8× speedup than the CVRL+ in [52]. We also find that although the BasicGEBD achieves fair detection performance with image-domain backbone, the BasicGEBD with CSN can achieve the detection performance of 82.5% without any sophisticated architecture designs, which is also superior to CVRL+, indicating the importance of conducting spatiotemporal modeling in the backbone models as we stated in Section 2.3. The higher performance of using CSN(R50)-L4* (81.1%) also confirms our findings. In Figure 7, we show that the family of evaluated EfficientGEBDs all achieve high efficiency in the experiments which demonstrate the effectiveness and efficiency of our design.

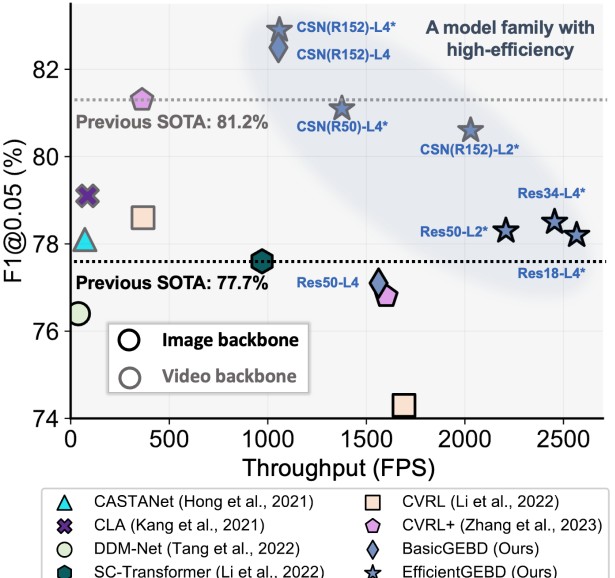

**Figure 7: The FPS *v.s* F1 score with evaluated GEBD methods on Kinetics-GEBD. FPS is measured using NVIDIA RTX 4090.**

## 3.3 Results on TAPOS

We also conduct experiments on the TAPOS [38] dataset in Table 3. The proposed EfficientGEBD also achieves the SOTA performance in terms of F1@0.05 (63.1%) and average (74.8%), respectively, which further proves the strong generalizability of our methods. More results and analyses on TAPOS are provided in supplementary materials. This verified the effectiveness of our method and our method can learn more robust feature presentation in different scenes.

## 3.4 Visualization

Based on the properties of the generic event boundaries, they can be specifically categorized into shot- (19%) and event-level (81%) boundaries. In this experiment, we further evaluate the performance of our methods in detecting different kinds of boundaries, where the pseudo recall[2] for each category is calculated and shown in Figure 8 (a). From the results, we see that shot-level boundaries

---

[2]Since the GEBD models only generate the boundary predictions regardless of their specific categories, the True Positive might not be true.

**Table 3: Comparison with others in terms of F1 score (%) on TAPOS with Rel.Dis. threshold from 0.05 to 0.5.**

| Method | F1 @ Rel. Dis. | | | | |
|---|---|---|---|---|---|
| | 0.05 | 0.1 | 0.3 | 0.5 | avg |
| ISBA [7] | 10.6 | 17.0 | 32.6 | 39.6 | 30.2 |
| TCN [26] | 23.7 | 31.2 | 34.4 | 34.8 | 64.0 |
| CTM [20] | 24.4 | 31.2 | 36.9 | 38.5 | 35.0 |
| TransParser [38] | 28.9 | 38.1 | 51.4 | 54.5 | 47.4 |
| PC [41] | 52.2 | 59.5 | 66.5 | 68.3 | 64.2 |
| Temporal Perceiver [46] | 55.2 | 66.3 | 76.5 | 78.8 | 73.2 |
| DDM-Net [47] | 60.4 | 68.1 | 75.3 | 76.7 | 72.8 |
| SC-Transformer [28] | 61.8 | 69.4 | 76.7 | 78.0 | 74.2 |
| **EfficientGEBD** (Res50-L3*) | **62.6** | **70.1** | **77.2** | **78.4** | **74.7** |
| **EfficientGEBD** (Res50-L4*) | **63.1** | **70.5** | **77.4** | **78.6** | **74.8** |

can be mostly effectively detected by the GEBD models. These event-level boundaries that have the largest number of samples (81%) are the samples that indeed make up the bulk of the miss-detection in GEBD tasks. The results also meet our hypothesis stated in Section 2.3, that the image backbone is more likely to suffer from the distraction issue when detecting event-level boundaries, resulting in a low detection recall (75.9%). While as using a video backbone effectively addresses the distraction issue, the detection recall can achieve an obvious improvement (80.1%) for these event-level boundaries.

We further provide qualitative results of shot- and event-level boundary detection on Kinetics-GEBD in Figure 8, from which we see that most predictions of our method are accurate. We also find that our method struggles with the detection when there are multiple small objects in the videos(shown in Figure 8 (d)). This meets our intuition since the changes of each object can be viewed as the boundary, which might distract the GEBD model and increase the complexity of detecting the event boundaries.

## 4 RELATED WORK

In video understanding fields, temporal detection tasks normally involve identifying clip-level instances from within untrimmed videos, such as shot boundary detection [14, 44, 48], temporal action segmentation [1, 8, 27], and temporal action localization [4, 35, 39, 51]. Initially introduced in [41], GEBD targets the localization of taxonomy-free moments, mirroring human perception of event boundaries according to recent cognitive science. These boundaries serve as crucial cues for a deeper understanding of long-form videos. Most existing research models the GEBD tasks as a binary classification problem for the input video clips. A line of research for GEBD tasks adopts a similar approach as described in [41] to partition lengthy videos into adjacent overlapping snippets, treating them as independent samples [13, 17, 23]. DDM-Net [47] further proposes to characterize the motion pattern with dense difference maps. There is also a series of works that conduct the entire video as a single input stream with continuous predictions [21, 28] and achieve higher efficiency. Moreover, Temporal Perceiver [46] builds the offline framework and achieves GEBD using a detection-based method.

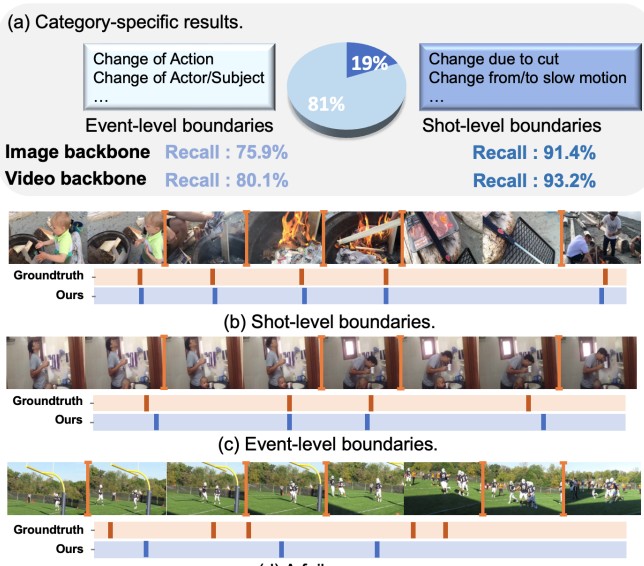

**Figure 8: Detection results on Kinetics-GEBD for (a) category-specific detection recalls. (b) shot-level changes, (c) event-level changes and (d) a failure case.**

Recent researchers have proposed to build the GEBD models in self-supervised or unsupervised manners. [24] proposes a recursive parsing algorithm based on the temporal self-similarity matrix to enhance local modeling. [37] builds a motion-aware GEBD method that uses a differentiable motion feature learning module to tackle spatial and temporal diversities. Furthermore, a parameter-free unsupervised GEBD detector is also proposed in [12], which conducts GEBD using optical flow without deep architectures.

It is more recent studies have also proposed to focus more on improving the efficiency of the GEBD models. In [13], a light-weighted GEBD detector is built based on transformer decoders. Moreover, an end-to-end compressed video representation learning (CVRL) for GEBD is proposed in [30, 52]. As recent studies [50, 54] have shown that using compressed video streams can speedup the inference, we believe that our method can also achieve higher efficiency when modified to allow the compressed video stream as input.

## 5 CONCLUSION

As an important pre-processing technique for long-form video understanding, the inefficiency of GEBD methods can largely affect the efficiency of long-form video processing, which prompts us to rethink the architecture design for efficient GEBD. In some ways, the performance and efficiency of the proposed BasicGEBD and EfficientGEBD are surprising while the model itself is not new, where most of the designs have been introduced in previous GEBD studies [28, 41, 47]. We also believe that combining other parallel techniques, such as self-supervised training [24, 45] or using compressed videos as inputs [29, 52], can further improve the performance of our method. We hope that the new results of this study will bring new intuitions for GEBD architecture design and new evaluation metrics that consider the model efficiency for GEBD tasks in video understanding fields.

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
