# OpenReview forum: "Rethinking the Architecture Design for Efficient Generic Event Boundary Detection"
_acmmm.org/ACMMM/2024/Conference — MM2024 Poster_

### Official Review · Reviewer_LfAU · 2024-05-07

**Rating:** 3
**Confidence:** 3

**Summary:**

i thank the authors for their hardwork and making the submission to acm multimedia.

**summary of paper**

this paper aims to tackle the task of generic event boundary detection. specifically, given a video, predict a temporal slice. the aim of the paper is to create an efficient architecture which performs well in terms of both accuracy and fps. (tables 2 and 3)

**Strengths:**

**strengths**
- fig 1 clearly manages to achieve the above goal.
- distraction issue: authors mention that image based backbones suffer from a distraction issue, i.e. the attention map obtained via gradcam focuses on body parts like neck of an actor instead of hands. (fig 6). it appears that video based models like csn focus on the parts like arms which actually move during the action. this lends an interesting perspective to the field, i.e. perhaps image based backbones are not sufficient to model temporal aspect.

**Limitations:**

**weaknesses**

the presented insights are interesting but they seem to be contradicting some well established behaviours in the field. i will be grateful if the authors can explain the following:

- effectiveness of image based backbones: in video-instance segmentation the sota models seem to be image based backbones i.e. autoregressive decoders in transformers. it does tracking really well (eg trackformer). one could even argue that image-based models are more efficient and do better than video-based models (eg vistr) in this case. this observation seems to conflict with the results of this paper. also, temporal information in image models is typically encoded by averaging the queries over time.

- fig 4: increasing the resnet model size results in lower performance i.e. resnet34 vs 8. why is that not obvious? is there any more subtle reason other than the fact that the net is overfitting. what was the length of the videos that the net processed?

- suppose a video model is being used eg, csn. how will it process a very long video? (>30 mins long). the memory will still remain an issue.

- the authors note  in lines 919:the model itself is not new, most of the designs have been introduced in previous gebd studies.
if then one accepts that the merits of this paper come from the analysis, perhaps the time will be well spent on explaining the distractor issue.

- are there only 2 datasets on generic event boundary detection? what about an additional task like temporal action localization.

- line 910-911: do the authors have any insight on how one could go about designing a net for processing compressed video streams.

- please add an additional discussion on the broader applications of this work. is this model useful in other domains also? can any of the insights gained be abstracted away to the field as a whole?
------
update 1 (may 07): due to above reasons i gave a borderline reject. i urge the authors to please resolve the above comments in rebuttal.

**Suitability:**

2

---

### Official Review · Reviewer_xtxZ · 2024-05-25

**Rating:** 5
**Confidence:** 3

**Summary:**

This paper reexamines the model design for the Generic Event Boundary Detection (GEBD) task and abstracts it into five components, each undergoing simplified design, resulting in BasicGEBD. Subsequently, DiffMixer is proposed, using a video-based CSN as the backbone, to obtain EfficientGEBD. The proposed model achieves state-of-the-art performance on two datasets and demonstrates significant improvements in terms of parameter count, FLOPs, and FPS compared to previous methods.

**Strengths:**

1. Through a thorough examination and extensive experiments on GEBD methods, this paper introduces BasicGEBD and EfficientGEBD, which not only enhance detection performance but also significantly reduce model computational costs. This work holds great value for applications in the field.
2. The writing is clear, and the experimental analysis is comprehensive, providing detailed insights. The figures and tables are visually intuitive.
3. The proposed DiffMixer, which captures both viewpoint changes and action variations, contributes to improving GEBD performance and exhibits a certain level of innovation.
4. The analysis of image backbone and video backbone provides valuable insights for selecting backbones in video understanding tasks.

**Limitations:**

1. Regarding the Kinetic-GEBD dataset, when using CSN as the backbone, EfficientGEBD shows limited improvement compared to BasicGEBD. This suggests that the proposed improvements in EfficientGEBD may not have a significant impact on the video backbone.
2. The results of BasicGEBD on the TAPOS dataset were not presented in the paper.
3. In line 329, it is mentioned that the training procedure is crucial for GEBD. For fair comparison, it is necessary to annotate or standardize the training procedures of previous methods.

**Suitability:**

3

---

### Official Review · Reviewer_n3PU · 2024-05-28

**Rating:** 5
**Confidence:** 2

**Summary:**

This paper discusses advancements in Generic Event Boundary Detection (GEBD), which segments videos into meaningful chunks, useful for tasks like video editing and summarization. The authors address the current problem by revisiting the model designs and making several key discoveries: a simple baseline model can perform well without complex designs; current models with image-domain backbones have redundant architecture; and a joint spatiotemporal modeling approach using video-domain backbones is more efficient. The resulting EfficientGEBD models show a performance boost and a practical speedup compared to previous methods.

**Strengths:**

1. The paper is well written and easy to follow.

2.The proposed  EfficientGEBD significantly outperforms previous state-of-the-art methods, showing up to 1.7% performance growth and 280% practical speedup under the same backbone choice.

3. The paper thoroughly analyzes different components of GEBD models, such as backbone redundancy and spatiotemporal learning, leading to a comprehensive understanding of the architecture.

**Limitations:**

1. Some pictures in the paper are not clearly explained. For example, what does the red star and the number next to it represent in Fig2? What do the different shades of blue represent? These need to be expressed in the caption.

2. The paper divides the GEBD method into these parts: backbone network; 2) encoder; 3) decoder; 4) fusion module. Are there any other architectures for the GEBD method?

**Suitability:**

3

---

### Meta-Review · Area_Chair_Uqd8 · 2024-07-04

**Recommendation:** Accept (Poster)
**Confidence:** 5

**Metareview:**

This paper addresses the task of Generic Event Boundary Detection (GEBD) by proposing efficient model designs that lead to improved performance and reduced computational costs. The authors introduce BasicGEBD and EfficientGEBD, focusing on a simplified model architecture and the use of video-based CSN backbones, respectively. The proposed models achieve state-of-the-art performance on two datasets. The reviewers generally agree that the paper is well-suited for multimedia/multimodal processing and provides significant contributions. The strengths, such as clear writing, comprehensive analysis, and innovative model designs, outweigh the limitations. However, the paper would benefit from addressing the identified weaknesses, particularly the clarity of figures, exploration of alternative architectures, and broader discussion on the model's applications.